# Impact of Amiodarone Therapy on the Ablation Outcome of Ventricular Tachycardia in Arrhythmogenic Right Ventricular Cardiomyopathy

**DOI:** 10.3390/jcm11247265

**Published:** 2022-12-07

**Authors:** Chin-Yu Lin, Fa-Po Chung, Nwe Nwe, Yu-Cheng Hsieh, Cheng-Hung Li, Yenn-Jiang Lin, Shih-Lin Chang, Li-Wei Lo, Yu-Feng Hu, Ta-Chuan Tuan, Tze-Fan Chao, Jo-Nan Liao, Ting-Yung Chang, Ling Kuo, Cheng-I Wu, Chih-Min Liu, Shin-Huei Liu, Wen-Han Cheng, Shih-Ann Chen

**Affiliations:** 1Heart Rhythm Center, Division of Cardiology, Department of Medicine, Taipei Veterans General Hospital, No. 201, Sec. 2, Shih-Pai Road, Taipei 112, Taiwan; 2Department of Medicine, National Yang-Ming Chiao-Tung University, No. 155, Sec. 2, Linong St., Taipei 112, Taiwan; 3Department of Cardiology, Yangon General Hospital, Yangon Q4HX+MH5, Myanmar; 4Cardiovascular Center, Taichung Veterans General Hospital, No. 1650, Taiwan Boulevard Sect. 4, Taichung City 40705, Taiwan

**Keywords:** amiodarone, arrhythmogenic right ventricular cardiomyopathy, abnormal electrograms, ventricular tachycardia, ablation

## Abstract

(1) Background: Catheter ablation (CA) is an accepted treatment option for drug-refractory ventricular tachycardia (VT) in patients with arrhythmogenic right ventricular cardiomyopathy (ARVC). This study investigates the effect of amiodarone on ablation outcomes in ARVC. (2) Methods: The study enrolled patients with ARVC undergoing CA of sustained VT. In all patients, substrate modification was performed to achieve non-inducible VT. The patients were categorized into two groups according to whether they had used amiodarone before CA. Baseline and electrophysiological characteristics, substrate, and outcomes were compared. (3) Results: A total of 72 ARVC patients were studied, including 29 (40.3%) “off” amiodarone and 43 (56.7%) “on” amiodarone. The scar area was similar between the two groups. Patients “off” amiodarone had smaller endocardial and epicardial areas with abnormal electrograms. Twenty of 43 patients (47.5%) “on” amiodarone discontinued it within 3 months after CA. During a mean follow-up period of 43.2 ± 29.5 months, higher VT recurrence was observed in patients “on” amiodarone. Patients “on” amiodarone who discontinued amiodarone after CA had a lower recurrence than those without. (4) Conclusions: Patients with ARVC “on” amiodarone before CA had distinct substrate characteristics and worse ablation outcomes than patients “off” amiodarone, especially in those who had used amiodarone continuously.

## 1. Introduction

Arrhythmogenic right ventricular cardiomyopathy (ARVC) is an inherited cardiomyopathy, characterized by fibro-fatty infiltration of the right ventricle owing to mutations in the desmosomal proteins [1]. Consequently, the heterogeneous substrates in the right ventricle are predisposed to delayed wavefront propagation and inhomogeneity of electrical conduction [2], which could contribute to ventricular tachycardia/fibrillation (VT/VF). Current guidelines recommend implanting an implantable cardioverter-defibrillator (ICD) in high-risk patients to prevent sudden cardiac death (SCD) or recurrence of fatal VT/VF in survivors [3]. Anti-arrhythmic drugs (AADs) are commonly used to prevent VT recurrences despite the implantation of an ICD. Amiodarone is the most common AAD used for the prevention of VT recurrences in patients with ARVC [4]. Amiodarone is, however, frequently limited by its ineffective reduction of recurrences of VT or well-known side effects [5]. Catheter ablation (CA) has therefore been used as an alternative procedure for patients who are intolerant to AADs or who have drug-refractory VT/VF in the ARVC [3]. However, the impact of amiodarone on substrate mapping and the subsequent effect on preventing VT recurrence after CA were not known.

Furthermore, a recent study demonstrated that concomitant amiodarone therapy may affect outcomes in patients with ischemic VT who are undergoing CA [6]. The above findings may be explained by changes in substrate characteristics and electrogram properties. It is unknown whether concomitant amiodarone therapy affects procedural outcomes in patients with ARVC. Accordingly, the aim of this study was to investigate the substrate characteristics and ablation outcomes between ARVC patients receiving CA with and without concomitant amiodarone therapy.

## 2. Materials and Methods

### 2.1. Study Population

From 2013 to 2020, we retrospectively screened patients with ARVC who had undergone successful CA of drug-refractory VT in three medical centers (Taipei Veterans General Hospital, Yangon General Hospital, and Taichung Veterans General Hospital). Based on the 2010 modified Task Force criteria, all patients met the diagnosis of definite ARVC [7]. The indications for catheter ablation included the following: (1) individuals with recurrent, sustained monomorphic VT refractory to antiarrhythmic drugs; and (2) symptomatic individuals with a high burden of ventricular ectopy and documented non-sustained VT refractory to antiarrhythmic drugs. Patients without documented sustained VT or successful CA were excluded. Patients had a minimum follow-up period of six months after CA. Written informed consent was obtained in all cases. The institutional review boards of each participating center approved the collection of data. From the electronic medical records, clinical data, such as baseline characteristics, 12 lead ECGs, 24 h Holter monitoring, transthoracic echocardiography, coronary angiography, and electrophysiological (EP)/ablation parameters, were collected for comparison. In the study, patients were divided into two groups based on their use of amiodarone before CA (“Off” amiodarone for at least 8 weeks before ablation and “On” amiodarone within 8 weeks). The use of amiodarone depends on physicians’ discretion or intolerable side effects.

### 2.2. Electrophysiological Study, Mapping, and Ablation

In previous publications, we described details of an electrophysiological study, substrate mapping, and ablation strategies [8]. After obtaining informed consent, we performed a standardized electrophysiological study on all patients under fasting and sedated conditions. Except for amiodarone, all anti-arrhythmic drugs (AADs) have been discontinued for at least five half-lives prior to CA [8]. The induction protocol and mapping strategy were described in the previous article [9]. Epicardial mapping/ablation is conducted in patients with failed endocardial ablation or in those who may have extensive epicardial involvement and limited abnormal substrates within the endocardium [8]. The RV endocardial bipolar scar and low voltage zone (LVZ) were defined as areas with peak-to-peak bipolar voltage < 0.5 mV and 0.5–1.5 mV, respectively, whereas the RV endocardial unipolar LVZ area was defined as an area with peak-to-peak unipolar voltage < 5.5 mV [10].

The RV epicardial bipolar scar and LVZ were defined as areas with a peak-to-peak bipolar voltage of <0.5 mV and 0.5–1 mV, respectively. The voltage maps were manually edited to avoid intracavitary points. An average bipolar or unipolar voltage was calculated. The scar, LVZ, and areas with abnormal electrograms (defined as either continuous, late, or fragmented potentials during sinus or paced rhythm) were assessed by using the standard surface area measurement tool on the navigation system [11]. When multiple areas of confluent low voltage were present, the aggregate area of the individual regions of interest was calculated. All patients received complete endocardial and/or epicardial substrate modification targeting the areas with abnormal electrograms. Successful ablation was defined as the non-inducibility of any VTs with or without isoproterenol [8].

### 2.3. Follow-Up and Recurrences of Ventricular Arrhythmia (VA)

Patients were followed up at the first, third, and sixth months after CA, as well as every three–six months thereafter. ICD interrogation, ECG, and Holter monitoring were performed. Recurrent VA, defined as sustained VT or VT/VF, requires ICD interventions [12].

### 2.4. Statistical Analysis

The continuous and categorical variables were expressed as mean ± standard deviation and percentage, respectively. The Student’s *t*-test was used to compare continuous variables, while chi-square tests with or without Yate’s correction for continuity or Fisher’s exact tests were used for categorical variables, as indicated. A multivariate Cox proportional hazards regression analysis was performed with variables with a *p*-value of less than 0.2 in the univariate analysis for a hazard ratio to predict the VT/VF recurrence. We considered a *p*-value of 0.05 to be statistically significant. The statistical analyses were conducted using the Statistical Package for the Social Sciences, Version 22.0 (IBM Corporation, Armonk, NY, USA).

## 3. Results

### 3.1. Study Population

#### 3.1.1. Patient Selection and Baseline Characteristics of ARVC Patients

In total, we screened 106 patients who were diagnosed with definite ARVC and had received successful CA of VA (Figure 1). The analysis included 72 patients (67.9%) (age 45.3 ± 14.5 years, 61.1% male) who received CA for clinically documented sustained VT with successful ablation. (Figure 1). Of these, 29 (40.3%) and 43 (59.7%) patients were classified as “off” and “on” amiodarone groups, respectively (Figure 1). The median (25–75%) duration of amiodarone use in patients “on” amiodarone was 11 (4–18) months before CA. The characteristics of these two groups are shown in Table 1. There were fewer men in patients “off” amiodarone (13 (44.8%) vs. 31 (72.1%); *p* = 0.027). Patients “off” amiodarone received more class Ic AAD medications (10 (34.5%) vs. 3 (7.0%); *p* = 0.004). In terms of the other parameters and Task Force criteria, there were no significant differences between the two groups.

#### 3.1.2. Endocardial and Epicardial Substrate Characteristics

Epicardial approach was performed in 47 (65.3%) patients, including 12 (41.4%) and 35 (81.4%) patients “off” and “on” amiodarone, respectively (*p* = 0.001). The characteristics of the RV substrate are shown in Table 2. There were no significant differences between the two groups regarding endocardial and epicardial bipolar voltage, unipolar voltage, and LVZ/scar area. The total activation time of the endocardium and epicardium was longer in patients “on” amiodarone than those in patients “off” amiodarone (158.2 ± 31.8 ms vs. 144.0 ± 32.7 ms, *p* < 0.001; 211.2 ± 26.0 ms vs. 183.6 ± 24.2 ms, *p* = 0.002). In addition, the areas of abnormal electrograms within the endocardium and epicardium were significantly larger in patients “on” amiodarone (Figure 2) than those in patients “off” amiodarone (Figure 3). (Endo: 16.8 ± 15.4 cm^2^ vs. 6.8 ± 5.0 cm^2^, *p* = 0.001; Epi: 28.2 ± 29.0 cm^2^ vs. 13.7 ± 21.4 cm^2^, *p* = 0.024). In the patients with amiodarone before CA, there was no significant difference in the substrate characteristics between patients with amiodarone durations of more than 6 months or not (Appendix A).

#### 3.1.3. Mapping of Ventricular Tachycardia and Catheter Ablation

During the electrophysiological study, 112 VTs (1.6 ± 0.8/patient) were induced, including 37 and 75 VTs in patients “off” and “on” amiodarone, respectively (*p* = 0.008). The mean cycle length of inducible VTs was 306.3 ± 66.1 ms. Of 112 inducible VTs, 61 VTs were mappable (14 in the “off” amiodarone group and 47 in the “on” amiodarone group, *p* = 0.007). Nine of the 61 mappable VT circuits were considered intramural because they lacked identifiable diastolic paths despite simultaneous epicardial and endocardial mapping. Three in the “off” amiodarone and six in the “on” amiodarone, (*p* = 0.731), all identified VT isthmuses and were correlated with areas with abnormal electrograms and abnormal substrates.

Each patient underwent ablation of the isthmus if mappable, and complete elimination of the endocardial and/or epicardial abnormal substrates. In patients “on” amiodarone, the total procedure duration and ablation time were significantly longer than in those “off” amiodarone (220.3 ± 43.9 min vs. 184.5 ± 51.1 min, *p* = 0.019, and 55.4 ± 37.7 min vs. 28.2 ± 24.4 min, *p* = 0.001, respectively). In all patients, non-inducibility of VT was achieved following CA. There were no procedural complications.

#### 3.1.4. Follow-Up

After a follow-up period of 43.2 ± 29.5 months, two patients (2.5%) died of non-cardiovascular causes (pneumonia: 2), and 24 (33.3%) experienced VT/VF recurrences. The incidence of VT/VF recurrence was significantly higher in patients “on” amiodarone than those “off” amiodarone (19/43 (44.2%) vs. 5/29 (17.2%), log-rank *p*-value = 0.011, Figure 4A). Among patients “on” amiodarone, the use of amiodarone was continued in 23 cases (53.5%) 3 months after CA. The incidence of VT/VF recurrence among patients on amiodarone before CA was still higher for those with continuous amiodarone use than for those without (14/23 (60.9%) vs. 5/20 (25.0%), log-rank *p*-value = 0.002, Figure 4B). Univariate and multivariate analyses (Table 3) showed that the use of amiodarone before CA (hazard ratio (HR): 3.04, 95% confidence interval (CI): 1.07–8.66, *p* = 0.038) is the only independent predictor of VT/VF recurrence.

## 4. Discussion

### 4.1. Main Findings

According to our study, amiodarone use before CA in patients with ARVC is associated with a longer activation time and a greater area of abnormal electrograms within the RV endocardium and epicardium. Despite the similar ablation strategy and endpoints (complete elimination of arrhythmogenic potential and non-inducible VT), VT recurrence was higher in patients “on” amiodarone before CA. Regardless of acute success, patients whose amiodarone use was discontinued 3 months after CA experienced fewer VT recurrences during subsequent follow-up. 

### 4.2. Amiodarone and the Abnormal Substrate

Amiodarone has been shown to prolong the total QRS duration and result in low amplitude signals in previous studies [13], which is consistent with the present findings that a longer activation time was observed in patients “on” amiodarone compared with those without. One of the main effects of chronic amiodarone therapy is the inhibition of potassium outward currents, which may prolong the action potential duration of the ventricular myocytes and delay conduction velocity [14]. Amiodarone may also increase the refractory periods of ischemic scar tissue and organize fractionated electrograms [15]. Luigi et al. reported that patients with ischemic VT “on” amiodarone had a smaller area of abnormal electrogram compared to those without [6]. These findings suggest that the use of amiodarone may mask “abnormal” electrical activity within the scar tissue in patients with ischemic VT. However, the impact of amiodarone on abnormal electrograms in patients with ARVC or other non-ischemic cardiomyopathies remains to be determined.

In contrast to the above-mentioned study regarding ischemic cardiomyopathy, we found that patients “on” amiodarone had larger areas of abnormal electrograms than those “off” amiodarone. Inconsistent findings as compared to ischemic substrates can be attributed to the following plausible explanations: (1) amiodarone prolongs conduction velocity heterogeneously in fatty-infiltrated myocytes and creates more abnormal electrograms in patients with ARVC; and (2) there is a diversity of disease progression and scarring in ARVCs, which could also contribute to nonuniform ablation outcomes. However, as a result of the study design, we cannot determine the cause-and-effect relationship between amiodarone use and the extent of abnormal electrogram extension in the present study. Further studies focusing on the effects of amiodarone on the diseased substrate in ARVC will be warranted. 

### 4.3. Amiodarone Use and Ablation Outcomes

In this study, we found that patients who were “on” amiodarone had higher recurrences of VT than those who were “off” amiodarone before CA. These findings may be explained by the following reasons: First, a greater extent of diseased substrates has been observed in amiodarone users, suggesting a worse disease status and/or faster disease progression in these individuals. Previously, we found that the progression of the disease could also cause the expansion of abnormal substrates, which was linked to VT recurrences [8]. The second issue is that, despite the larger area with abnormal electrograms, there were still some arrhythmogenic substrates that were hidden after amiodarone use, as seen in ischemic substrates. As a result, substrate modification could not be performed completely. Moreover, amiodarone could possess good antiarrhythmic properties and prevent ventricular arrhythmias from being induced after CA [16]. Nonetheless, the possibility of this is negated by the findings that these patients were refractory to amiodarone before CA. Furthermore, the pro-arrhythmic effect was reported with a relatively low incidence but was not negligible [4,17]. Amiodarone might prolong the QT interval and cause polymorphic ventricular tachycardia (i.e., torsade de pointes), especially in patients with drug-drug interactions [18]. The above hypothesis might be indirectly supported by the observation that patients with continuous use of amiodarone after ablation experience VT recurrences at higher rates. Amiodarone, however, causes proarrhythmia at a rate of less than one percent per year [19]. A previous study demonstrated that long-term amiodarone use did not increase mortality or hospitalization [20]. Further investigation is needed to determine the exact reason for the higher incidence of VT recurrence in ARVC patients with amiodarone use before CA. 

### 4.4. Clinical Implications

To the best of our knowledge, this is the first study to investigate the impact of amiodarone use before CA in patients with ARVC. Amiodarone use before CA can result in distinct substrate characteristics, such as longer activation times and larger areas with abnormal electrograms and was associated with worse ablation outcomes. According to the above findings, we suggest the physician withdraw amiodarone before CA if the patient can tolerate it. Despite the standard ablation strategies and endpoints, VT recurrences were still higher in patients without discontinuation, reflecting the importance of alternative strategies to assess the ablation endpoints in these patients who could not withdraw the amiodarone. Future studies are warranted to address this aspect. 

### 4.5. Limitations

There were some limitations in the present study. This is a retrospective study in which only patients who had successfully undergone ablation were included. The prescription of amiodarone was based on the clinical physician’s discretion. In patients with ARVC before CA, it is difficult to standardize the use of AADs. Furthermore, the use of amiodarone after CA was also at the discretion of the physicians, which may have affected the present findings. In this retrospective study, it is difficult to determine whether the amiodarone or the need for amiodarone affects substrate characteristics and recurrence. Nevertheless, the bias can be minimized by multivariate analysis and collaboration between referral centers. Additionally, epicardial approaches were not performed in all patients. However, even with the above-mentioned differences, there are more patients “on” amiodarone who are undergoing epicardial ablation to achieve the same outcome. In this regard, we concluded that epicardial ablation was not the cause of the differences between the two groups. A prospective study is warranted to validate the impact of amiodarone on the ablation outcome in patients with definite ARVC.

## 5. Conclusions

Despite similar ablation strategies and endpoints, amiodarone use before CA may be associated with a higher rate of VT recurrence in patients with ARVC. Higher VT recurrences were observed in ARVC patients who continued to take amiodarone despite the acute success.

## Figures and Tables

**Figure 1 jcm-11-07265-f001:**
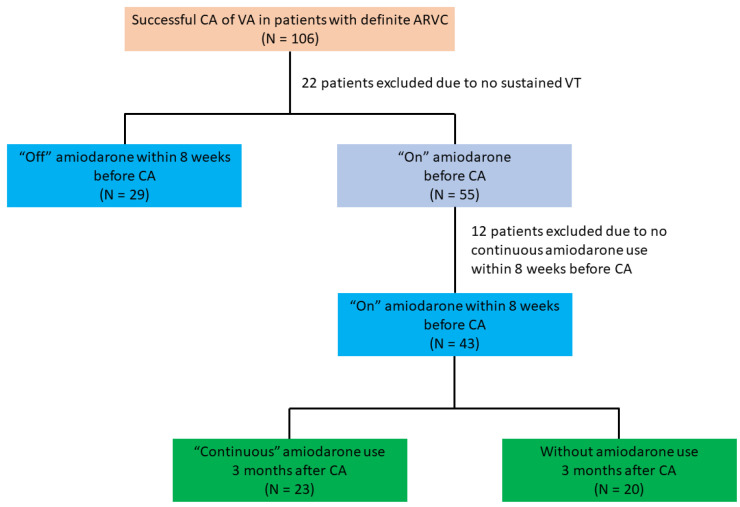
Patient selection in the present study. Patients with ARVC receiving successful CA for ventricular arrhythmia were screened. We enrolled ARVC patients with clinically documented sustained VT, and patients were further categorized into two groups based on their use of amiodarone within the 8 weeks before CA. ARVC = arrhythmogenic right ventricular cardiomyopathy; CA = catheter ablation; VT = ventricular tachycardia.

**Figure 2 jcm-11-07265-f002:**
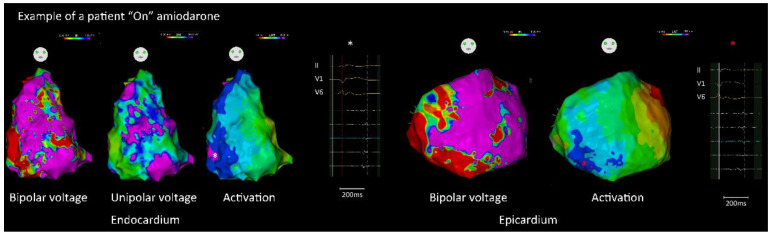
An example of electrophysiological and substrate characteristics in a patient “on” amiodarone before CA. Patients with ARVC “on” amiodarone had a longer total activation time and larger areas with abnormal electrograms. The (**left panel**) showed an endocardial voltage map and activation map with local abnormal electrograms. The (**right panel**) showed an epicardial voltage map and activation map with local abnormal electrograms (* indicated the location of corresponding signal). “See text for details”.

**Figure 3 jcm-11-07265-f003:**
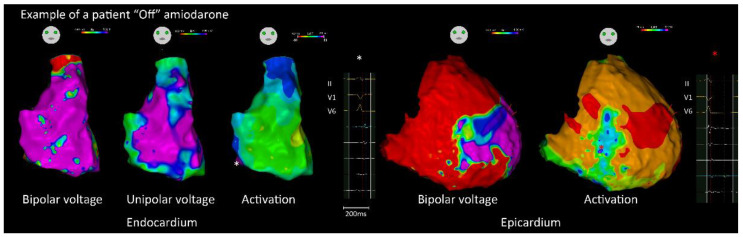
An example of electrophysiological and substrate characteristics in a patient “off” amiodarone before CA. The (**left panel**) showed an endocardial voltage map and activation map with local abnormal electrograms. The (**right panel**) showed an epicardial voltage map and activation map with local abnormal electrograms (* indicated the location of corresponding signal). “See text for details”.

**Figure 4 jcm-11-07265-f004:**
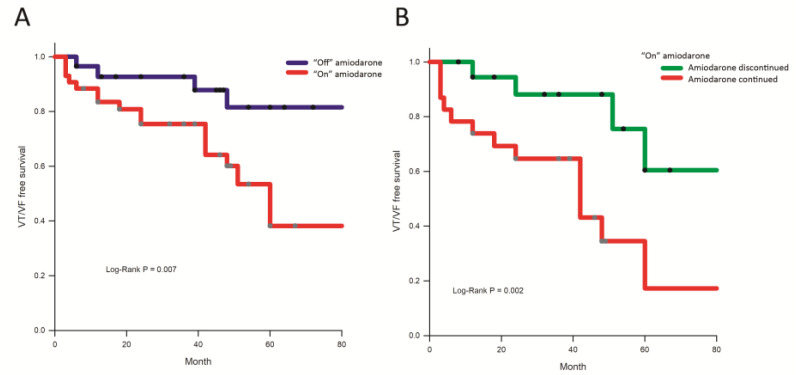
Kaplan–Meier curve for ablation outcomes during follow-up. (**A**) The Kaplan–Meier curve showed the recurrence-free survival in patients “on” and “off” amiodarone. Higher VT/VF recurrences were observed in patients “on” amiodarone than in those “off” amiodarone. (**B**) The Kaplan–Meier curve demonstrated the VT/VF recurrence-free survival in the patients with or without continuous amiodarone use after the successful ablation in patients “on” amiodarone before CA.

**Table 1 jcm-11-07265-t001:** Comparison of baseline characteristics between patients with ARVC “On” and “Off” amiodarone before CA.

	“Off” Amiodarone	“On” Amiodarone	*p*-Value
	(N = 29)	(N = 43)	
Baseline characteristics			
Age	43.0 ± 13.1	46.9 ± 15.4	0.260
Sex (Men, %)	13 (44.8%)	31 (72.1%)	0.027
Hypertension	7 (24.1%)	14 (32.6%)	0.598
Diabetes mellitus	2 (6.9%)	2 (4.7%)	0.999
LVEF (%)	57.2 ± 8.1	53.5 ± 8.7	0.081
RVEF (%)	41.0 ± 9.7	38.0 ± 11.5	0.240
Medication			
Amiodarone dose before CA (mg)	0	232.6 ± 97.9	
Beta blocker before CA	21 (72.4%)	29 (67.4%)	0.795
Class Ic AAD before CA	10 (34.5%)	3 (7.0%)	0.004
Class Ib AAD before CA	7 (24.1%)	4 (9.3%)	0.105
“Continued” amiodarone before discharge	0	38 (88.4%)	
“Continued” amiodarone 3 months after CA	0 (0.0%)	23 (53.5%)	
Amiodarone dose after CA (mg)	0	223.9 ± 106.5	
Task Force criteria *			
Structural abnormalities			
Major	9 (31.0%)	25 (58.1%)	0.076
Minor	15 (51.7%)	14 (32.6%)	
Fibrofatty replacement			
Major	7 (24.1%)	9 (20.9%)	0.483
Minor	2 (6.9%)	13 (30.2%)	
Depolarization abnormalities			
Major	2 (6.9%)	9 (20.9%)	0.138
Minor	26 (89.7%)	30 (69.8%)	
Repolarization abnormalities			
Major	9 (27.3%)	7 (15.2%)	0.421
Minor	13 (39.4%)	21 (45.7%)	
Ventricular arrhythmias			
Major	10 (34.5%)	19 (44.2%)	0.469
Minor	19 (65.5%)	24 (55.8%)	
Family history			
Major	5 (17.2%)	10 (23.3%)	0.568
Minor	0 (0.0%)	1 (2.3%)	

* According to the 2010 revised Task Force criteria (9). AAD, anti-arrhythmic drug; ARVC, arrhythmogenic right ventricular cardiomyopathy; CA, catheter ablation; LVEF, left ventricular ejection fraction; RVEF, right ventricular ejection fraction.

**Table 2 jcm-11-07265-t002:** Comparison of baseline characteristics between patients with ARVC “On” and “Off” amiodarone before ablation.

	“Off” Amiodarone	“On” Amiodarone	*p*-Value
	(N = 29)	(N = 43)	
VT morphologies	1.3 ± 0.6	1.7 ± 0.9	0.014
VT cycle length	309.2 ± 54.7	304.5 ± 72.5	0.774
Procedure time (minutes)	184.5 ± 51.1	220.3 ± 43.9	0.019
Ablation time (minutes)	28.2 ± 24.4	55.4 ± 37.7	0.001
RV endocardium			
Mapping points	600.7 ± 528.5	850.0 ± 666.8	0.096
Bipolar voltage *	2.1 ± 0.6	1.9 ± 0.7	0.142
Unipolar voltage *	5.3 ± 0.7	4.9 ± 1.4	0.253
Total activation time (ms)	144.0 ± 32.7	158.2 ± 31.8	<0.001
Bipolar low voltage zone (cm^2^)	28.4 ±17.1	34.7± 27.4	0.275
Bipolar low voltage zone, %	13.5 ± 7.6	15.4 ± 10.7	0.415
Bipolar scar (cm^2^)	14.2 ± 12.5	17.4 ± 13.6	0.317
Bipolar scar, %	6.8 ± 5.3	8.0 ± 5.5	0.341
Unipolar low voltage zone (cm^2^)	55.1 ± 36.3	65.3 ± 27.9	0.183
Unipolar low voltage zone, %	22.6 ± 11.7	25.8 ±10.2	0.215
Area with abnormal electrograms	6.8 ± 5.0	16.8 ± 15.4	0.001
RV epicardium	(N = 12)	(N = 35)	
Mapping points	1495.8 ± 950.9	2076.8 ± 1776.9	0.270
Averaged bipolar voltage (mV) *	1.3 ± 0.5	1.3 ± 0.8	0.919
Total activation time (ms)	183.6 ± 24.2	211.2 ± 26.0	0.002
Bipolar low voltage zone (cm^2^)	95.0 ± 54.8	111.5 ± 55.8	0.378
Bipolar low voltage zone, %	36.0 ± 22.2	36.3 ± 20.4	0.968
Bipolar scar (cm^2^)	53.4 ± 43.7	56.6 ± 34.2	0.797
Bipolar scar, %	19.0 ± 14.9	19.1 ± 11.4	0.842
Area with abnormal electrograms	13.7 ± 21.4	28.2 ± 29.0	0.024

* The average of bipolar or unipolar median voltage. ARVC, arrhythmogenic right ventricular cardiomyopathy; RV, right ventricular; VT, ventricular tachycardia.

**Table 3 jcm-11-07265-t003:** Univariate and multivariate analyses of VT/VF recurrence after catheter ablation.

	Univariate Analysis		Multivariate Analysis	
	*p*-Value	HR (95% CI)	*p*-Value	HR (95% CI)
Age (year)	0.995	1.00 (0.97–1.03)	-	-
Sex (male)	0.051	2.56 (0.99–6.56)	0.605	1.38 (0.41–4.68)
Hypertension	0.062	2.21 (0.96–5.10)	0.665	1.24 (0.47–3.22)
Diabetes mellitus	0.758	1.38 (0.18–10.46)	-	-
LVEF (%)	0.642	0.99 (0.95–1.04)	-	-
RVEF (%)	0.049	0.96 (0.93–1.00)	0.494	0.98 (0.94–1.03)
Amiodarone before CA	0.013	3.54 (1.31–9.60)	0.038	3.04 (1.07–8.66)
Amiodarone > 6 months	0.283	1.56 (0.69–3.54)		
Task Force score	0.550	1.08 (0.84–1.38)	-	-
Total activation time (RV endo)	0.009	1.02 (1.00–1.03)	0.385	1.03 (0.96–1.12)
Bipolar scar (%, RV endo)	0.283	1.04 (0.97–1.12)	-	-
Epicardial approach	0.413	1.44 (0.60–3.46)	-	-

CA = catheter ablation; CI = confidence interval; HR = hazard ratio; LVEF = left ventricular ejection fraction; RV = right ventricle; RVEF = RV ejection fraction.

## Data Availability

The authors confirm that the data supporting the findings of this study are available within the article.

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
