# Peer review of "Impact of Amiodarone Therapy on the Ablation Outcome of Ventricular Tachycardia in Arrhythmogenic Right Ventricular Cardiomyopathy"

_jcm, 2022, doi:10.3390/jcm11247265_

Round 1

Reviewer 1 Report

The article is novel and well-written.

Author Response

# 1 The article is novel and well-written.

Response:

Thanks for your comment!

Reviewer 2 Report

I read the paper with great interest.  However, some points in discussion are rather based on  hypothesis than study results. Firstly, I guess the usage of amiodarone before CA was based on clinical arrhythmia,  so the patients with more complex arhythmias were treated with amiodarone. The fact could explain the results of the EP study.  The Authors properly discussed that issue in the paper. The same may explain the follow up results after CA. O wonder if Authors suggest amiodarone withdrawal before CA? It seems to be reasonable on the basis of their results and discussion. Could Authors answer my question in the discussion section?

The exclusion criteria are not clear. I do not understand why 22 pts were excluded. Were they ablated due to another indications than VT? Please clarify that in order to improve the quality of the very good paper.

The results are very interesting and of great clinical importance.

Author Response

Responses to Reviewer 2

#1. I read the paper with great interest.  However, some points in discussion are rather based on  hypothesis than study results. Firstly, I guess the usage of amiodarone before CA was based on clinical arrhythmia,  so the patients with more complex arhythmias were treated with amiodarone. The fact could explain the results of the EP study.  The Authors properly discussed that issue in the paper. The same may explain the follow up results after CA. O wonder if Authors suggest amiodarone withdrawal before CA? It seems to be reasonable on the basis of their results and discussion. Could Authors answer my question in the discussion section?

Reply:

Thanks for your great comment.

In our present study, amiodarone use before CA can result in distinct substrate characteristics, such as longer activa-tion times and larger areas with abnormal electrograms, and was associated with worse ablation outcomes. According to the above findings, we suggest physician withdraw amiodarone before CA if patient could tolerate. Despite the standard ablation strategies and endpoints, VT recurrences were still higher in patients without discontinuation, reflecting the importance of alternative strategies to assess the ablation endpoints in these patients who couldn’t withdraw the amiodarone. Future studies are warranted to address this aspect.

Please see the revised discussion in the section “4.4. Clinical implications (line 256-260)” as follows, “

Amiodarone use before CA can result in distinct substrate characteristics, such as longer activa-tion times and larger areas with abnormal electrograms, and was associated with worse ablation outcomes. According to the above findings, we suggest physician withdraw amiodarone before CA if patient could tolerate. Despite the standard ablation strategies and endpoints, VT recur-rences were still higher in patients without discontinuation, reflecting the importance of alterna-tive strategies to assess the ablation endpoints in these patients who couldn’t withdraw the amiodarone. Future studies are warranted to address this aspect.

#2. The exclusion criteria are not clear. I do not understand why 22 pts were excluded. Were they ablated due to another indications than VT? Please clarify that in order to improve the quality of the very good paper.

Reply:

Thanks for your great comment.

The indications for catheter ablation included the following: (1) individuals with recurrent sustained monomorphic VT refractory to antiarrhythmic drugs, and (2) symptomatic individuals with a high burden of ventricular ectopy and documented nonsustained VT refractory to antiarrhythmic drugs.

In this present study, we aimed to enrolled the patients with sustained VT refractory to antiarrhythmic drugs. The 22 patients were excluded because there was no documented sustained VT.

We revised the section of “2.1 Study population (line 68-71)” as follows,

The indications for catheter ablation included the following: (1) individuals with recurrent sus-tained monomorphic VT refractory to antiarrhythmic drugs, and (2) symptomatic individuals with a high burden of ventricular ectopy and documented nonsustained VT refractory to anti-arrhythmic drugs. Patients without documented sustained VT or successful CA were excluded.

Reviewer 3 Report

Thank you very much for the opportunity of reviewing this paper. Authors investigated how amiodarone could change electrophysiological properties of the tissue and affect outcome after effective catheter ablation of ventricular arrhythmias.

The major concern about this paper is that it is really hard to determine whether it is the amiodarone  or the need for amiodarone to affect substrate characteristics.  Authors reported that patients receiving amiodarone before ablation had a near double presence of major structural abnormalities and more morphologies of ventricular tachycardias. It makes sense because the greater the extent of the disease the greater the likelihood of having arrhythmias and the greater the chance to have relapses after ablation.  

Moreover, you can't speak of association between amiodarone use and substrate findings or outcome without running  multivariable models. You should test the administration of amiodarone as an independent variable and correct  for disease extent, age, sex and other potential cofounders.

Author Response

#1 The major concern about this paper is that it is really hard to determine whether it is the amiodarone  or the need for amiodarone to affect substrate characteristics.  Authors reported that patients receiving amiodarone before ablation had a near double presence of major structural abnormalities and more morphologies of ventricular tachycardias. It makes sense because the greater the extent of the disease the greater the likelihood of having arrhythmias and the greater the chance to have relapses after ablation. Moreover, you can't speak of association between amiodarone use and substrate findings or outcome without running  multivariable models. You should test the administration of amiodarone as an independent variable and correct  for disease extent, age, sex and other potential cofounders.

Reply:

Thanks for your great comments.

We agreed with your comments that it is really hard to determine whether the amiodarone or the need for amiodarone affects substrate characteristics and recurrence in this retrospective study.

We performed the multivariate analysis with CA regression in the new Table 3 accordingly. After multivariate analysis, the use of amiodarone before CA independently predicted VT/VF recurrence after successful ablation.

Please see the revised manuscript as follows, “

Section 2.4 Statistical analysis (line 113-115)

A multivariate Cox proportional hazards regression analysis was performed with variables with a P-value of less than 0.2 in the univariate analysis for a hazard ratio to predict the VA recurrence.

Section 3.1.4. Follow-up (line 167-169)

Univariate and multivariate analyses (Table 3) showed that the use of amiodarone before CA (hazard ratio [HR]: 3.04, 95% confidence interval [CI]: 1.07-8.66, P = 0.038) is the only independent predictor of VT/VF recurrence.

Section 4.5. Limitations (line 265 – 269)

Furthermore, the use of amiodarone after CA was also at the discretion of the physicians, which may have affected the present findings. It is really hard to determine whether it is the amiodarone or the need for amiodarone to affect substrate characteristics and recurrence in this retrospective study. Nevertheless, the bias can be minimized by multivariate analysis and collaboration between referral centers.

Please see the new Table 3 as follows, “

Table 3. Univariate and multivariate analyses of VT/VF recurrence after catheter ablation

Univariate analysis

Multivariate analysis

p value

HR (95% CI)

p value

HR (95% CI)

Age (year)

0.995

1.00 (0.97-1.03)

-

-

Sex (male)

0.051

2.56 (0.99-6.56)

0.605

1.38 (0.41-4.68)

Hypertension

0.062

2.21 (0.96-5.10)

0.665

1.24 (0.47-3.22)

Diabetes mellitus

0.758

1.38 (0.18-10.46)

-

-

LVEF (%)

0.642

0.99 (0.95-1.04)

-

-

RVEF (%)

0.049

0.96 (0.93-1.00)

0.494

0.98(0.94-1.03)

Amiodarone before CA

0.013

3.54 (1.31-9.60)

0.038

3.04 (1.07-8.66)

Task Force score

0.550

1.08 (0.84-1.38)

-

-

Total activation time (RV endo)

0.009

1.02 (1.00-1.03)

0.385

1.03 (0.96-1.12)

Bipolar scar (%, RV endo)

0.283

1.04 (0.97-1.12)

-

-

Epicardial approach

0.413

1.44 (0.60-3.46)

-

-

CA = catheter ablation; CI = confidence interval; HR = hazard ratio; LVEF = left ventricular ejection fraction; RV = right ventricle; RVEF = RV ejection fraction; VF = ventricular fibrillation; VT = ventricular tachycardia

Round 2

Reviewer 3 Report

Thank you very much for the effort spent in addressing my comments. You did a good job by providing both a univariable and multivariable analysis. It could be of interest including in the model also the time spent on amiodarone before the CA. If you could demonstrate that the longer  the time on drug and the greater are the changes of the substrate you could provide more convincing elements about the effective role of amiodarone  rather than the need of drug.

Author Response

Responses to Reviewer 3

#1 Thank you very much for the effort spent in addressing my comments. You did a good job by providing both a univariable and multivariable analysis. It could be of interest including in the model also the time spent on amiodarone before the CA. If you could demonstrate that the longer  the time on drug and the greater are the changes of the substrate you could provide more convincing elements about the effective role of amiodarone  rather than the need of drug.

Reply:

Thanks for your great comments.

The median duration of the amiodarone usage in the “On” amiodarone group is 6 months (4-16 months). We further divided the “On” amiodarone group into two groups with amiodarone use for more than six months or not.

We added the parameter “amiodarone more than 6 months” for the analysis in the revised Table 3.

Table 3. Univariate and multivariate analyses of VT/VF recurrence after catheter ablation

Univariate analysis

Multivariate analysis

p value

HR (95% CI)

p value

HR (95% CI)

Age (year)

0.995

1.00 (0.97-1.03)

-

-

Sex (male)

0.051

2.56 (0.99-6.56)

0.605

1.38 (0.41-4.68)

Hypertension

0.062

2.21 (0.96-5.10)

0.665

1.24 (0.47-3.22)

Diabetes mellitus

0.758

1.38 (0.18-10.46)

-

-

LVEF (%)

0.642

0.99 (0.95-1.04)

-

-

RVEF (%)

0.049

0.96 (0.93-1.00)

0.494

0.98(0.94-1.03)

Amiodarone before CA

0.013

3.54 (1.31-9.60)

0.038

3.04 (1.07-8.66)

Amiodarone > 6 months

0.283

        1.56 (0.69-3.54)     

Task Force score

0.550

1.08 (0.84-1.38)

-

-

Total activation time (RV endo)

0.009

1.02 (1.00-1.03)

0.385

1.03 (0.96-1.12)

Bipolar scar (%, RV endo)

0.283

1.04 (0.97-1.12)

-

-

Epicardial approach

0.413

1.44 (0.60-3.46)

-

-

CA = catheter ablation; CI = confidence interval; HR = hazard ratio; LVEF = left ventricular ejection fraction; RV = right ventricle; RVEF = RV ejection fraction; VF = ventricular fibrillation; VT = ventricular tachycardia

We also analyzed the substrate parameter between the patient with amiodarone used for more than 6 months or not. The substrate characteristics were similar between long and short amiodarone administration before ablation. Please check the revised manuscript line 142-144 and the new Supplemental Table 1 as follows,

Supplemental Table  1. Comparison of substrate characteristics between patients with different periods on amiodarone before ablation

Amiodarone > 6 months

Amiodarone more than 8 weeks and ≤ 6 months

p-value

(N = 21)

(N = 22)

RV endocardium

   Bipolar voltage*

1.8 ± 0.6

2.0 ± 0.7

0.222

Unipolar voltage*

4.9 ±1.7

4.9 ± 1.2

0.999

Total activation time (ms)

167.4 ± 27.2

161.3 ±34.2

0.496

Bipolar low voltage zone (cm2)

40.4 ±32.3

29.2± 20.9

0.185

Bipolar low voltage zone, %

17.3± 12.9

13.6± 8.0

0.262

Bipolar scar (cm2)

19.5 ± 15.8

15.4 ± 11.2

0.332

Bipolar scar, %

8.8 ± 6.6

7.3 ± 4.3

0.385

Unipolar low voltage zone (cm2)

65.2 ± 34.2

65.4 ± 21.2

0.977

Unipolar low voltage zone, %

26.6 ± 12.8

26.6 ±7.9

0.999

Area with abnormal electrograms

17.7 ± 17.4

15.8 ± 13.7

0.692

RV epicardium

N=16

N=19

   Bipolar voltage*

1.5 ± 1.0

1.1 ± 0.4

0.152

Total activation time (ms)

212.3 ± 31.6

210.3 ±21.0

0.825

Bipolar low voltage zone (cm2)

97.6± 53.9

123.2 ±56.0

0.180

Bipolar low voltage zone, %

33.1± 22.3

38.9 ±18.8

0.410

Bipolar scar (cm2)

51.1 ± 34.0

61.2 ± 34.6

0.390

Bipolar scar, %

17.4 ± 13.4

18.6 ± 9.7

0.756

Area with abnormal electrograms

24.7 ± 30.1

31.8 ± 28.1

0.434

*The average of bipolar or unipolar median voltage.

ARVC, arrhythmogenic right ventricular cardiomyopathy; RV, right ventricular; VT, ventricular tachycardia
